

# Real-time quintic Hermite interpolation for robot trajectory execution

Morten Lind

Production Technology, SINTEF Manufacturing, Trondheim, Trøndelag, Norway

## ABSTRACT

This paper presents a real-time joint trajectory interpolation system for the purpose of frequency scaling the low cycle time of a robot controller, allowing a Python application to real-time control the robot at a moderate cycle time. Interpolation is based on quintic Hermite piece-wise splines. The splines are calculated in real-time, in a piecewise manner between the high-level, long cycle time trajectory points, while sampling of these splines at an appropriate, shorter cycle time for the real-time requirement of the lower-level system. The principle is usable in general, and the specific implementation presented is for control of the Panda robot from Franka Emika. Tracking delay analysis is presented based on a cosine trajectory. A simple test application has been implemented, demonstrating real-time feeding of a pre-calculated trajectory for cutting with a knife. Estimated forces on the robot wrist are recorded during cutting and presented in the paper.

## INTRODUCTION

Reacting instantaneously to perceived changes in the environment is an important feature of modern robotics, recently for safe human-robot cooperation, but also more traditionally for force-based control and visual servoing applications. The setup for this includes a sensor system for perceiving the environment, a control model for deciding how to react, and a control interface to the mechanical system. This must all be real-time integrated into a sensor-based robot control application.

A real-time interface to the robot arm, through the robot controller, is a necessity. Newer robot controllers are getting increasingly open towards real-time trajectory feeding. An early example of a supported mechanism was realized in the KUKA controllers with the Robot Sensor Interface (RSI) software package, which used a cycle time of 12 ms in version 2 and 4 ms in version 3; version 3 maintained the legacy 12 ms cycle time as an option. Results were presented by *Lind, Schrimpf & Ulleberg (2010)*.

The Universal Robots controllers can be controlled in various, official ways through the standard controller process. Methods and results are reported by *Andersen (2015)*. Older versions of the Universal Robots controllers provided an unofficial C API, with very low tracking delay, as presented by *Lind, Schrimpf & Ulleberg (2010)*. More accurate methods for measuring and classifying types of delays was presented by *Andersen et al. (2015)*.

Corresponding author
Morten Lind, morten.lind@sintef.no

Besides standard interfaces to robot controllers, some controllers can be hacked for gaining access at the lower controller levels. *Kröger & Wahl (2010)* directly accessed the frequency inverters of a Stäubli RX60 controller and was able to perform servo-level control in 10 kHZ. *Schrimpf (2013)* injected a single-board computer in the USB-link between the higher and lower level controller components in a Nachi AX10 controller, enabling joint-level control in 100 Hz.

Developing sensor-based, real-time robot control applications is challenging. More so when the target programming platforms, in general, are exclusively available in compiled languages such as C++. For those situations where the real-time performance is not too critical, also called soft real-time requirements, it will increase the development efficiency, if a higher level, general programming language, such as Python, can work as the target platform. The main motivation for the presented work is to enable such a high level, general, soft real-time programming platform, in this case Python, for developing advanced control application; such as the DeepMPC framework for cutting of food products described by *Lenz, Knepper & Saxena (2015)* and the real-time grasping control described by *Morrison, Leitner & Corke (2018)*.

From experiments and experience, a pure Python-based framework, exploiting NumPy for numerical calculations, manages to calculate kinematics for a typical industrial robot system with six to nine joints in 10 ms, with room for also doing some sensor processing and general data accounting, using a contemporary PC (*Lind, Tingelstad & Schrimpf, 2012*). This calculation time strongly depends on the amount for sensor processing and data accounting, and on the computing power on the CPU and computer system on which the software is deployed. On a modern, high-end PC of today the results presented by *Lind, Tingelstad & Schrimpf (2012)* may well be possible with a 5 ms cycle time. Due to the "Global Interpreter Lock" in the Python interpreter, multi-threading in a single Python process is not utilizing the multiple cores of modern CPUs, and computationally heavy applications should be distributed over several processes. *Eggen & Eggen (2019)* presents experiments and results aimed at determining when it is advantageous to process-distribute a Python application. *Schrimpf, Lind & Mathisen (2013)* presented a time analysis for various data paths in a distributed, real-time, sensor-based robot application implemented and deployed with Python and ROS.

The seven-axis Panda robot from Franka Emika can be controlled in real-time through an Ethernet UDP connection. The real-time control interface requires a cycle time of 1 ms. Several other robot controllers provide real-time interfaces for trajectory feeding at rates higher than 100 Hz. *Mihelj et al. (2012)* mentions official and research interfaces for real-time control through available controllers for KUKA, Stäubli, Yaskawa Motoman, and Comau robots. These examples may indicate a slow but general tendency for such joint-level, low-latency interfaces to become of more general availability among robot controllers in the future.

The cycle time requirement of 1 ms of the Franka controller, and even the 4 ms interface to newer KUKA controllers with RSI version 3, poses a computational and real-time challenge for Python-based real-time trajectory feeding application. It is a computation time challenge, since *pure* Python computations in general is slower than C++ code by an

order of magnitude or more. This, however, is mitigated somewhat when using efficient scientific computation packages such as NumPy and SciPy. It is a real-time challenge, since the wake-up latency on the PC gets an additional contribution through the Python interpreter.

This paper shortly sketches a prototype solution addressing these challenges by presenting a joint-level intermediary motion service over Ethernet that respects the 1 ms real-time obligation towards the Franka controller, while providing an interface to the Python motion application requiring a more moderate cycle time; e.g., 10 ms. The motion service requires a cyclic real-time response from the motion application while obeying the similar requirement from the Franka controller. This allows for the motion application to make instantaneous changes and correction to the generated trajectory it emits. A positive side effect of this motion service is that it is re-connectable, since it detects the disconnection from, or a failure to comply in, the Python application, whereby it takes over the real-time obligation towards the Franka controller; and thus the low-level control loop is maintained.

The implementation of the presented solution is based on piece-wise quintic Hermite splines with an option to limit velocity, acceleration and jerk. The current implementation was developed for being fed a position trajectory from the motion application. However, velocities and accelerations are derived over the position trajectory and used in the interpolation, and can also be fed from the motion application. The presented solution can be classified as Type V in the scheme by *Kröger & Wahl (2010)*.

The choice of piece-wise Hermite spline as the fundamental computational object for trajectory representation was made due to its adequate parameterization and domain: Piece-wise Hermite splines are parameterized by clamping the direct motion quantities, such as position, velocity, acceleration, and jerk, at the end-points of each segment.

## SYSTEM AND METHODS

The minimal system that have been set up for experimentation and testing is based on the seven-axis Panda robot from Franka Emika (https://franka.de/). An intermediate motion service is performing the interpolation from a lower frequency loop to the required rate of the high frequency loop for positions, and optionally velocities and accelerations, for all joints. A motion application may then connect to the low-frequency interface of this motion service, and thus control the robot arm in a moderate real-time frequency. This section gives an overview of the active entities in the experiment system and the Hermite spline calculus.

### The Panda Robot and the Franka Controller

At the lowest level the robot arm servos are controlled via a proprietary interface by the "Franka Controller" (FC). The "Franka Control Interface" (https://frankaemika.github.io/docs/libfranka.html) (FCI) is a software addon for the FC which enables real-time trajectory feeding, synchronized with reading the state of the robot; including joint torques and estimated external forces.

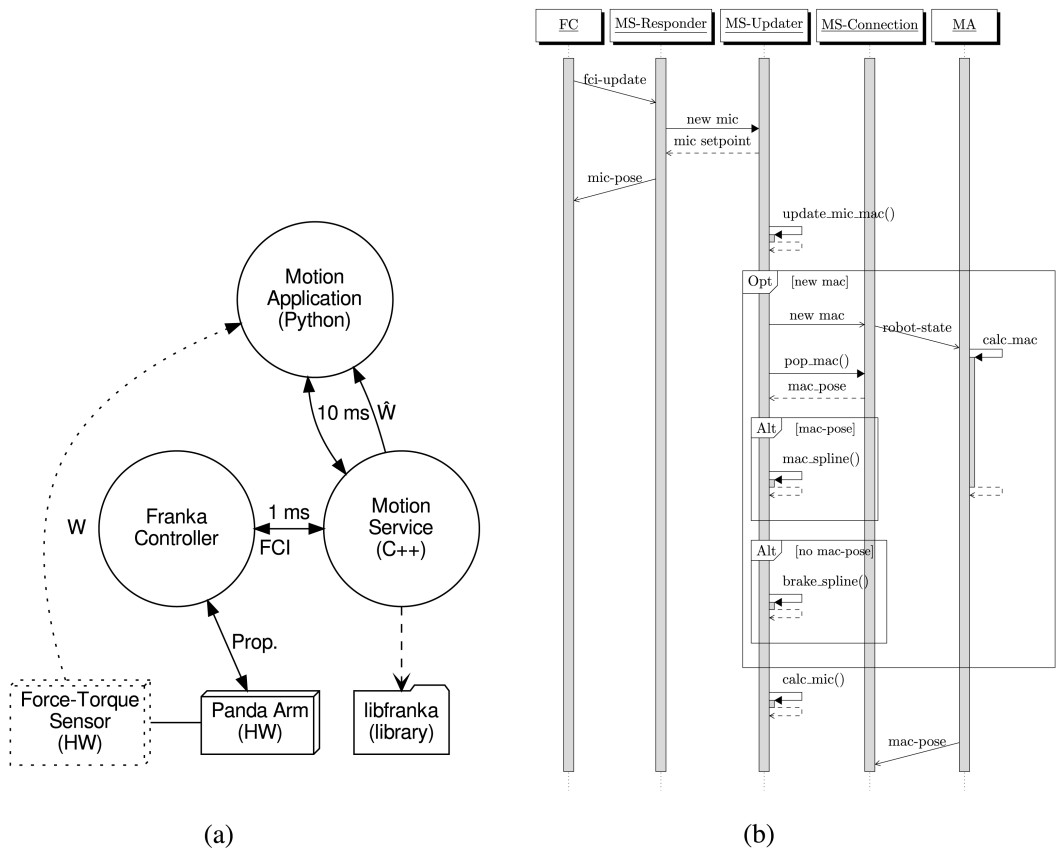

(a)                                                    (b)

**Figure 1   Interaction and sequence diagrams for the proposed system.** (A) Rough structure of processes and entities with their connections. [Image credit: Morten Lind, 2020]. (B) Sequence diagram for the synchronization among the three threads in MS, the FA and the MA. Image credit: Morten Lind, 2020.

When the FCI is installed on the FC, the freely available C++ library "libfranka"(https://github.com/frankaemika/libfranka) can be used to operate the robot arm at a cycle time of 1 ms. The mechanism is to hand a callback function to libfranka when starting the control loop. This callback is then invoked every 1 ms with the status of the robot arm, expecting to be returned within less than one cycle. This cycle time is dubbed the *micro* cycle time.

In Figure 1A FC is modelled as a process controlling the Panda arm, running on the Franka Controller PC, i.e., it is born with its own node. The FCI is exposed over an Ethernet UDP socket, on an IP address configured in the FC. Thus the trajectory feeding is performed from a different node; e.g., a user-supplied PC. libfranka provides various methods of control, ranging from operation (Cartesian) space control of the tool flange to joint torque control. This paper does not go into detail with libfranka and the presented work only uses the joint position control mode.

## Motion service and application

For the purpose of obeying the micro cycle real-time requirement the "Motion Service" (MS) has been developed and implemented. As illustrated in Fig. 1A, MS links with libfranka for setting up the callback communicating with the FC over FCI, and starting the control loop.

The MS is running in its own process, on its own node, separate from the FC. The "Motion Application" (MA) process may run on the same computational node as MS. For connection responsiveness, it is better to have the MS and the MA run on the same node. However, for dedicating more computing power to the MA, leaving more computational resources for sensor analysis and control algorithms, it could be running on a dedicated node. The presented architecture, where the MS is exposing its interface via an Ethernet UDP socket, is flexible in this regard.

Figure 1A illustrates how an optional force-torque sensor may be added. Such a force-torque sensor would typically be mechanically mounted between the wrist of the robot arm and the robot tool for measuring the wrench, $W$, on the tool. A force-torque sensor is not used in the presented setup and experiments. Instead the estimated wrench, $\hat{W}$ from the robot dynamics is used. This estimate is obtained as a feature from libfranka and provided by the MS to the MA.

## Software mechanisms

For understanding the general workings of the MS, this section gives a brief overview of the central mechanistic design.

Figure 1B shows a sequence diagram illustrating the central, operative synchronization and responsibility of the three threads of the MS code. In addition the network-remote entities FC and MA are modelled as threads with remote message interaction to the MS threads.

In addition to this central, operative interaction the connection thread in the MS has a complex logic for maintaining control of the connection from an MA. This will not be treated here. Even though the connection management, which enables the FC-MS interaction to be kept alive over repeated MA-sessions, the main focus of this paper is to describe the interpolation in the MS.

Also not illustrated in Fig. 1B is the setup and initialization procedures of the MS, which establishes the real-time loop with the FC and prepares for receiving connections from an MA. Like the connection maintenance, this is also important, but the details are not in the focus of this paper.

A micro cycle is initiated by the callback from the FC to the Responder thread in the MS, conveying the robot arm status. The Responder thread is only responsible for conveying this trigger and status data to the Updater thread, and for returning the prepared next micro setpoint to the FC obtained from the Updater thread in response. Upon receiving the new micro state and returning the next micro setpoint, Updater thread is triggering itself to calculate the micro setpoint for the next micro cycle.

When the Updater thread is triggered to prepare the micro setpoint for the next micro cycle, it first updates its micro and macro cycle accounting, which determines whether a

new macro cycle is to be started. The current macro cycle is valid for the next micro cycle if the current spline domain is valid for the time of the next micro cycle. In that case, the current spline is used to prepare the next micro setpoint.

If a new macro cycle is to be started, the Updater triggers the Connection thread, and the next macro setpoint is retrieved from it. If a valid new macro pose is retrieved, a new spline based on this is set up, and used for micro setpoint calculation. If not, due to late response or disconnect from the MA, a braking spline is generated. The severity of this braking spline depends on whether the missing macro setpoint was due to lateness or a broken connection. In the former case, only light braking will be generated, since it is expected that the MA will resume its responsiveness. In the latter case full deceleration braking is generated since the robot arm then must come to a full stop as fast as possible.

The Connection thread is, during operation in an MA-session, responsible for the interaction with the MA. It is triggered by the Updater thread when a new macro cycle is started, which it propagates on to the MA. After triggering the MA, it expects to receive a new macro setpoint some time later, before the end of the newly started macro cycle. When a macro setpoint is received from the MA, a flag is set to tell that a new macro setpoint is received. This flag is checked by the Updater thread, and is used to determine, if the MA was within its deadline. The Updater clears the flag, preparing it for receiving the next macro setpoint from the MA.

Timing-wise, when the macro cycle is between macro setpoint $i$ and $i+1$, a received macro setpoint is stored for use as the $i+3$'th macro setpoint; i.e., one extra macro setpoint in the stream is retained. This delay is introduced to be able to correctly estimation the acceleration to target when splining toward $i+2$, calculated using the central acceleration estimator over the macro setpoints $i+1$, $i+2$, and $i+3$. The same principle is used for the velocity at macro setpoint $i+2$, which results in the average velocity on the segments $i+1 \rightarrow i+2$ and $i+2 \rightarrow i+3$.

The calculus of motion quantities at the macro trajectory points is based on the stream of joint positions received from the MA, which are assumed to arrive at regular times separated by the macro cycle time, $\delta t_{\mathrm{mac}}$. This stream of macro joint position vectors, $\{\mathbf{p}_i\}_i$, is the fundamental input data from the MA to the MS. Over these, the central discrete estimators of velocities and accelerations may be concisely expressed as

$$\mathbf{p}_{i+2}^{(1)} = 0.5 \left[ \frac{\mathbf{p}_{i+2} - \mathbf{p}_{i+1}}{\delta t_{\mathrm{mac}}} + \frac{\mathbf{p}_{i+3} - \mathbf{p}_{i+2}}{\delta t_{\mathrm{mac}}} \right] \tag{1}$$

$$\mathbf{p}_{i+2}^{(2)} = \frac{\mathbf{p}_{i+3} - 2\mathbf{p}_{i+2} + \mathbf{p}_{i+1}}{\delta t_{\mathrm{mac}}^2} \tag{2}$$

These estimated first and second derivatives of the macro joint trajectory are used for calculating the splines, which are then sampled to generate the micro trajectory positions.

## Hermite splines

Hermite spline and Hermite interpolation are named in honour of the 19th century French mathematician Charles Hermite. General treatment on multi-node Hermite splines of arbitrary order may be found in several publications and textbooks. E.g., *Spitzbart (1960)*

focused on a general formulation for arbitrary order, while *Krogh (1970)* focused on efficient computation of interpolation and numerical differentiation with continuous derivative. However, course notes by *Finn (2004)* introduces an elegant formulation for the basis polynomials for cubic and quintic Hermite interpolation. Both have been implemented in the MS. The formalism can be concisely written, with the same enumeration of the basis functions, as

$$\tilde{p}_n(u) = \sum_{i=0}^{(n-1)/2} p_0^{(i)} H_i^n(u) + p_1^{(i)} H_{n-i}^n(u) \quad \text{for } u \in [0;1] \tag{3}$$

In Eq. (3) we assume an underlying trajectory $p(u)$ of which we know $p_\tau^{(i)} = p^{(i)}(\tau)$ for $i \in \{0..n-1\}$ and $\tau \in \{0,1\}$, where $p^{(i)}$ is the $i$'th derivative of $p$. The basis functions for Hermite interpolation to order $n$ are denoted $\{H_i^n\}_{i\in\{0..n\}}$. For cubic ($n=3$) and quintic ($n=5$) Hermite interpolation the basis functions can be found listed in *Finn (2004)*. For completenes the coefficients are listed in matrix notation in Eqs. (4) and (5).

$$\mathbf{H}^3 = \begin{bmatrix} 1 & 0 & -3 & 2 \\ 0 & 1 & -2 & 1 \\ 0 & 0 & -1 & 1 \\ 0 & 0 & 3 & -2 \end{bmatrix} \tag{4}$$

$$\mathbf{H}^5 = \begin{bmatrix} 1 & 0 & 0 & -10 & 15 & -6 \\ 0 & 1 & 0 & -6 & 8 & -3 \\ 0 & 0 & \frac{1}{2} & -\frac{3}{2} & \frac{3}{2} & -\frac{1}{2} \\ 0 & 0 & 0 & \frac{1}{2} & -1 & \frac{1}{2} \\ 0 & 0 & 0 & -4 & 7 & -3 \\ 0 & 0 & 0 & 10 & -15 & 6 \end{bmatrix} \tag{5}$$

In an online or real-time system a change of variable is necessary for Eq. (3) to be applicable. When the system at run-time establishes a new interpolation segment it takes the form $[t_0; t_1]$ with $\delta t_{\text{mac}} = t_1 - t_0$. Introducing the substitution on the interval $u(t) = \frac{t-t_0}{\delta t_{\text{mac}}}$, and letting $p_\tau^{(i)} = p^{(i)}(\tau)$ for $i \in \{0..n-1\}$ and $\tau \in \{t_0, t_1\}$, the directly applicable version of Eq. (3) is

$$\tilde{p}_n(t) = \sum_{i=0}^{(n-1)/2} (\delta t_{\text{mac}})^i \left( p_{t_0}^{(i)} H_i^n(u(t)) + p_{t_1}^{(i)} H_{n-i}^n(u(t)) \right) \quad \text{for } t \in [t_0; t_1] \tag{6}$$

Equation (6) has been implemented in the MS code base for both cubic and quintic Hermite splines, and can be used based on joint velocities and accelerations from the macroscopic trajectory which is fed from the MA.

It is noteworthy that a piece-wise cubic Hermite spline is a $C^1$-smooth function and a piece-wise quintic Hermite spline is a $C^2$-smooth function. With respect to generating motion trajectories, the great difference between these is the jerk; the third derivative of the position trajectory. Many motion controllers require being fed a limited-jerk trajectory, which is fulfilled by a $C^2$-trajectory, whereas a $C^1$-trajectory exhibits infinite jerk.

# EXPERIMENTS AND RESULTS

During development a cosine position trajectory generator have been used extensively. This section presents some results from such cosine trajectories. A cosine trajectory is used for testing, since it starts at zero speed, is cyclic, and is infinitely smooth ($C^\infty$). In addition to the testing by a cosine position trajectory, preliminary experiments have been performed using the force-torque estimation from the FC with a Python-based framework, aimed at integrating force-feedback in a knife-cutting application. The cutting experiment does not utilize the real-time possibility of the Motion Service, but only serves to demonstrate its soundness from a simple application perspective.

## Cosine joint motion

A simple cosine motion is generated and the motion of a single joint is observed. This will first and foremost give insight into the tracking delay in the system. The presented results defines the tracking delay from the MC perspective as the time passed from receiving a macro setpoint from the MA until a status packet from the FC shows that the joint position have been achieved.

Figure 2B show a selected time range for the position trajectory of the moving joint. The time range have been selected to easily inspect the delays from the MA over the "Franka MS" (FMS), and to the report back from the FC. The observed delay between the MA and the FMS is approximately 20 ms with the delay from the FMS to the reported FC position is another approximate 20 ms. In total, for the trajectory execution we can estimate a 40 ms delay.

It is evident that there is an inherent tracking delay of approximately 20 ms in the FC. This is affected by various filters and compliance parameters which are set at the initialization of the control loop through the FCI. Thus a lower inherent tracking delay may possible with more tight control parameters. Particularly the experiments presented here have been using a value of 10.0 Hz for the parameter `cutoff_frequency` in libfranka. This parameter controls, according to the documentation of libfranka, a first order low-pass filter. The reason for choosing a low cutoff frequency is observable as a "blackout" of two to three micro cycles near $t = 3.185$s in Fig. 2. Such blackouts occurs generally every second, but not regularly. The reason for these blackouts have not been clarified, but they probably arise from running the system on a full desktop PC; i.e., there have been no stripping of services or other running processes on the PC to increase the real-time responsiveness.

Execution of a simple cosine motion in one joint also allows the observation of the difference between using cubic and quintic Hermite splines. To this end, classes have been implemented in the FMS code base for both of these. Figure 3 shows accelerations obtained by discrete derivation of the MA and FMS trajectories for the observed joint. Figure 3A shows the acceleration in a trajectory in a session where the FMS runs with cubic Hermite interpolation and Fig. 3B shows the same type of cosine trajectory from a session using quintic Hermite interpolation.

The cubic Hermite interpolation acceleration curve in Fig. 3A is clearly discontinuous, which is consistent with its $C^1$-property. Correspondingly the quintic Hermite interpolation acceleration curve in Fig. 3B is clearly continuous, which is consistent with its $C^2$-property.

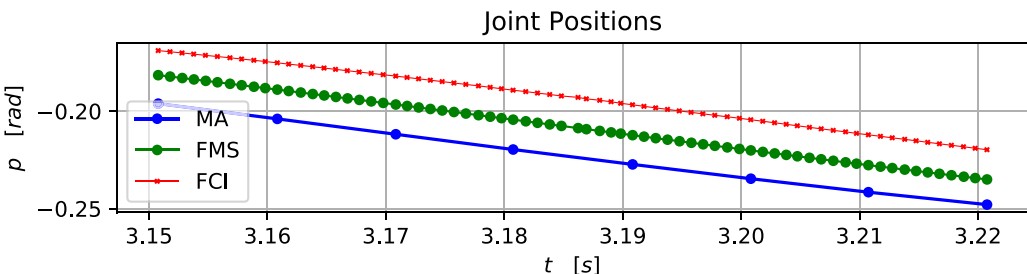

**Figure 2** A selected time range of the positions in a cosine trajectory. Image credit: Morten Lind, 2020.

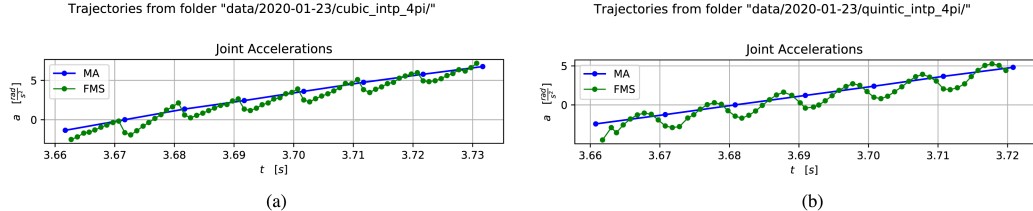

**Figure 3** **Comparison of accelerations in the generated trajectories from a cosine motion using cubic and quintic Hermite interpolation.** (A) Acceleration for cubic interpolation. Image credit: Morten Lind, 2020. (B) Acceleration for quintic interpolation. Image credit: Morten Lind, 2020.

## Simple cutting

A simple application for a cutting experiment has been set up. This is an application where the robot tool is a knife and the task is to cut through a presented object. By simple cutting is meant moving the tool according to a pre-calculated trajectory; in contrast to sensor-based adaptive cutting where sensor input is used in the motion generation loop.

The application in this experiment does not exploit the ability to make real-time generation of, or corrections to, the commanded trajectory. The purpose of the experiment is to do a simple demonstration of the robustness of the MS implementation and to get an indication of the reliability and stability of the wrist wrench estimation obtained from libfranka.

The minimalistic setup used in the experiment is illustrated in Fig. 4. Figure 4A shows a photo of the knife held in the robot gripper and a test object; in this case a stick of EPS strapped to a bracket.

Figure 4B illustrates the geometric setup of the knife. The knife shaft is held clamped in the gripper, and thus defines the commanded knife directions for cutting $\hat{c}_c$, which is perpendicular to the edge, and shearing $\hat{s}_c$, which is parallel to the edge. The actual cutting and shearing directions are defined by the knife edge at the point of interaction. These are illustrated as $\hat{c}_a$ and $\hat{s}_a$, respectively. Interpretation of the data for the recorded forces in a

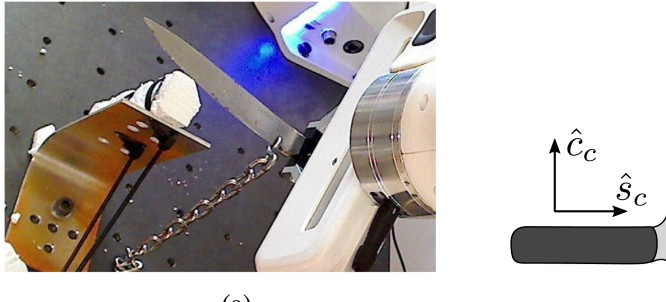

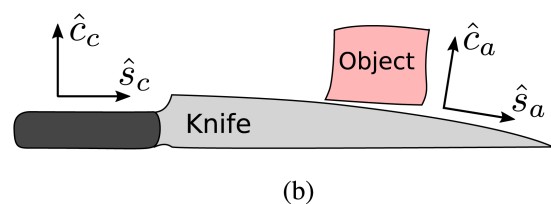

(a)  (b)

**Figure 4** **The setup of the knife and object.** The cutting and shearing directions used for commanding the motion is aligned with the knife, whereas the actual cutting and shearing directions depend on the angle of the knife blade in the region of interaction. (A) Photo of the setup of robot, knife, and object. Photo credit: Morten Lind, 2020. (B) Illustration of controlled and actual cutting and shearing directions. Image credit: Morten Lind, 2020.

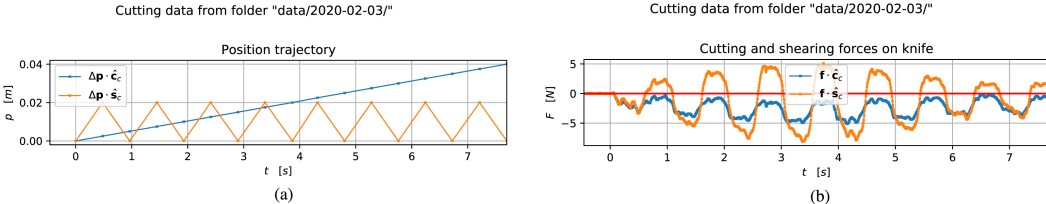

(a)  (b)

**Figure 5** **Position and force recording sampled through a cutting process. Image credit: Morten Lind, 2020.** (A) Commanded trajectory of the knife along the $\hat{c}_c$ and $\hat{s}_c$ directions. Image credit: Morten Lind, 2020. (B) Forces acting on the knife along the $\hat{c}_c$ and $\hat{s}_c$ directions. Image credit: Morten Lind, 2020.

cutting experiment is deeply dependent of the recognition of the relation between these commanded and actual directions.

A cutting experiment is executed by moving the knife in a cyclic series of strokes, forward strokes in the positive $\hat{s}_c$ direction and backward strokes in the negative $\hat{s}_c$ direction, while progressing steadily in the positive $\hat{c}_c$ direction. The motion of the knife happens thus in a plane spanned by the $\hat{c}_c$ and $\hat{s}_c$. The commanded position trajectory is shown in Fig. 5A. The origin of the positions is where the knife is held at the start of the cutting process.

The corresponding forces on the knife along the $\hat{c}_c$ and $\hat{s}_c$ directions, based on the estimated wrench obtained from libfranka in the FMS, is seen in Fig. 5B. In an ideal experiment one would expect a constant, negative cutting force in the $\hat{c}_c$ direction and a square wave shearing force, symmetric around zero, in the $\hat{s}_c$ direction. Both of these would be smoothly modulated in size by the initially increasing, and later decreasing, interaction of the knife with the object as it passes through it.

There are a couple of deviations from this ideal, qualitative behaviour. Most noteworthy is the non-smoothness of the direct cutting force. Next is the asymmetry around zero of the shear force. Both of these are easily understood by the illustration of the actual cutting and shearing directions observed in Fig. 4B.

## DISCUSSION

The current major challenge regards the blackouts in the communication between MA, FMS, and FCI. For a reliable and stable system, this must be addressed by real-time hardening of the computing environment on which the application runs. However, this is fairly unrelated to the soundness of the presented approach and reference implementation.

Particularly for knife-cutting applications, the discussion about a negative bias on the cutting and shearing forces, observed in Fig. 5B, due to the deviation from the commanded directions, is interesting for future development. It shows that an edge-object-interaction observer should be developed, which will certainly be important for correct execution of the cutting process. The presented implementation will be used for future experiments in two directions involving real-time force feedback to the trajectory generation. Firstly in the direction of explicitly modelling the cutting process, based on such work as that of *Reyssat et al. (2012)*. Secondly, on the more implicit learning approach to cutting presented by *Lenz, Knepper & Saxena (2015)*.

Another application area, for which the presented implementation is intended, is that of grasping of unknown or variable objects. The vision-based servo control involved with "closed-loop grasping" is executing at a fairly low frequency; in the order of 5 Hz according to *Morrison, Leitner & Corke (2018)*. Thus, whereas it is expected that the required frequency and tracking delay will be a challenge for a Python-based application when it comes to real-time control of cutting, this is not expected to be a challenge for real-time grasping applications.

It was originally under consideration to use the Reflexxes (http://reflexxes.ws) library developed by Torsten Kröger for the underlying interpolation mathematics in the presented Motion Service process. However, the Reflexxes version under a free software license, LGPL V3.0, only provides $C^1$-smooth trajectories. For generating $C^2$-smooth trajectories, a commercial license of Reflexxes must be purchased. The presented software is intended to be free (GPL), and thus the Reflexxes library was not chosen.

## CONCLUSION

In summary, this work has

- established a re-connectable real-time motion service via the Franka Control Interface for the Panda robot;
- characterized the real-time trajectory execution performance by illustrating the tracking delay;
- demonstrated that real-time motion application from Python is possible;
- indicated the feasibility of using the external force-torque estimator provided by libfranka at the real-time application level.

## ACKNOWLEDGEMENTS

Thanks to Ekrem Misimi at SINTEF Ocean for general guidance and for supporting the work.

### Funding

The underlying work and the writing of this paper was funded by the Norwegian Research Council (https://www.forskningsradet.no/en/) in the project "Innovative and Flexible Food Processing Technology in Norway" with project number 255596 (https://prosjektbanken.forskningsradet.no/#/project/NFR/255596). The funders had no role in study design, data collection and analysis, decision to publish, or preparation of the manuscript.

### Grant Disclosures

The following grant information was disclosed by the author:
Norwegian Research Council.
Innovative and Flexible Food Processing Technology in Norway:  255596.

### Competing Interests

Morten Lind is an employee of SINTEF Manufacturing AS. The author declares that he has no competing interests.

### Author Contributions

- Morten Lind conceived and designed the experiments, performed the experiments, analyzed the data, performed the computation work, prepared figures and/or tables, authored or reviewed drafts of the paper, and approved the final draft.

### Data Availability

Code is publicly available in the "Franka Motion Service" repository at GitLab: https://gitlab.com/SINTEFManufacturing/franka_motion_service.git. Data are publicly available in the "FMS Test Data" repository at https://gitlab.com/SINTEFManufacturing/fms-test-data.git.

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
