# Peer review of "Real-time quintic Hermite interpolation for robot trajectory execution"

_PeerJ Computer Science, doi:10.7717/peerj-cs.304_

## Round 0.1 · original submission · Minor Revisions

Besides all other revisions, please concentrate on experimental design and validation of your model, since both Reviewers rise those aspects.

Reviewer 1 ·

Basic reporting

This paper is well-written. The number of references is relatively small.

Experimental design

The authors only consider one joint to perform experiment. In future, three or more joints are suggested to consider, since it would have more strict requirements on real-time。

Validity of the findings

The experimental results demonstrated that real-time motion application from Python is possible.

Additional comments

This paper investigates a real-time joint trajectory interpolation system for the purpose of frequency scaling the low cycle time of a robot controller. A re-connectable real-time motion service is established. And the experimental results demonstrate that real-time motion application from Python is possible. This paper is well-written. And the following comments could be considered.

1. In Introduction, it is suggested to add some statements about contributions of this paper.

2. It is suggested to add some related references.

3. This paper only considers one joint to perform experiments. In future, a robot with three or more joints is suggested to consider, since it would have more strict requirements on real-time。

·

Basic reporting

a. A central term in the article is Real-time. The author should more clearly distinguish between when he is talking about Computer Science Real-time (a deterministic execution period with no relation to latency) and a, in lack of a better expression, “linguistic” real-time (the notion of events happening “right now” with no/low latency)
b. I wonder if the results of Lind et al. from 2012 still holds with respect to arguing that Python is inadequate in calculating kinematics fast enough on a contemporary computer.
c. The claim at line 62-64 that Numpy and SciPy is a magnitude slower than C++ is unsupported. As both Python packages are implemented in C I doubt if this holds.
d. I am missing a discussion regarding the choice of interpolation method. Why use Hermite interpolation instead of I.e. regular Newtonian interpolation. No reference to other papers using Hermite interpolation is mentioned in the introduction. Is it a novel idea or a commonly used method for joint trajectory interpolation?
e. I generally think more effort should be put into explaining the chosen interpolation method to make it more of a stand-alone work and to make it easier to follow along.

Typos/ grammar:
Fig 1b: The newly calculated mic-pose should be returned to from Updater to Responder.
Proof read lines 103, 125, 150

Experimental design

a. Why is a cosine motion chosen, when only a (almost) linear path of the trajectory is shown?
b. Why not use a TCP connection to the controller instead of UDP (mentioned at line 108) to avoid the “blackouts” described at line 191. The first line of the discussion states that these blackouts are currently the biggest challenge, but TCP should ensure that all packages are received.
c. Why is the absolute acceleration values shown in figure 3a different from those in figure 3b when the author at line 199 states that it is the same trajectory?
d. Is the position error fed back to the Motion Service?
i. If yes, how is the tracking actually performing compared to the commanded trajectory shown in figure 5a?
ii. If not, how is the cutting action relevant for the contributions of the article?

Validity of the findings

No comment

Additional comments

Generally a sound work. The chosen cutting application and sudden shift of focus from trajectory tracking to force estimation is a bit confusing.

---

## Round 0.2 · accepted · Accept

The revision process is now finished and independent Reviewers recommend your paper for further processing, therefore I am pleased to forward your paper to the editorial office with a positive recommendation.

Reviewer 1 ·

Basic reporting

No comment

Experimental design

No comment

Validity of the findings

No comment

Additional comments

The version of the paper can be accepted.

·

Basic reporting

no comment

Experimental design

no comment

Validity of the findings

no comment

Additional comments

After reviewing the changes and the author's rebuttal letter, I have no further comments or suggestions for improvements